# Assessment of Multi-Use Offshore Platforms: Structure Classification and Design Challenges

**Walid M. Nassar \***[ID]**, Olimpo Anaya-Lara, Khaled H. Ahmed, David Campos-Gaona**[ID]
**and Mohamed Elgenedy**

Department of Electronic & Electrical Engineering, Faculty of Engineering, University of Strathclyde,
Royal College Building, 204 George St, Glasgow G1 1XW, UK; olimpo.anaya-lara@strath.ac.uk (O.A.-L.);
khaled.ahmed@strath.ac.uk (K.H.A.); d.campos-gaona@strath.ac.uk (D.C.-G.);
mohamed.elgenedy@strath.ac.uk (M.E.)
\* Correspondence: walid-mohamady-hassan-nassar@strath.ac.uk

**Abstract:** As the world continues to experience problems including a lack of seafood and high energy demands, this paper provides an assessment for integrated multi-use offshore platforms (MUPs) as a step towards exploiting open seawater in a sustainable way to harvest food and energy. The paper begins with background about MUPs, including information regarding what an MUP is and why it is used. The potential energy technologies that can be involved in an offshore platform are introduced while addressing similar applications all over the world. The paper presents the state of the art of MUP structures on the light of EU-funded programs. An MUP would have a positive impact on various marine activities such as tourism, aquaculture, transport, oil and gas and leisure. However, there are concerns about the negative impact of MUPs on the marine environment and ecosystem. Building an MUP with 100% renewable energy resources is still a challenge because a large storage capacity must be considered with a well-designed control system. However, marine bio-mass would play a vital role in reducing battery size and improving power supply reliability. Direct Current (DC) systems have never been considered for offshore platforms, but they could be a better alternative as a simpler control system that requires with lower costs, has lower distribution losses, and has an increased system efficiency, so studying the feasibility of using DC systems for MUPs is required.

**Keywords:** multi-use platform; aquaculture; offshore; energy

## 1. Introduction

Water covers 71% of the surface of the earth, and oceans hold around 96.5% of this water [1]. As a result of gradually increasing global warming emissions and increasing of populations all over the world, there is a high tendency for more sustainable activities to cut these emissions and provide food in a sustainable way. Oceans, as an alternative, have a lot of opportunities for the energy and food sectors. To exploit the ocean's resources, we must go offshore and use platforms that are suitable for different kinds of activities. These platforms are known as multi-use platforms (MUPs) or multi-purpose platforms (MPPs). An MUP includes sustainable components and activities, which are discussed in detail in Section 2, such as wind, solar, floating tidal, wave and ocean thermal energy. Additionally, aquaculture should be designed in a sustainable way as per the revised European Union Commission Fisheries Policy (CFP) and its strategic guidelines for the sustainable development of the European Union aquaculture, which are intended to guide the development of aquaculture in Europe such that "it can contribute to the overall objective of filling the gap between the European Union (Member Organization) consumption and production of seafood in a way that is environmentally, socially and economically sustainable." These objectives exist in addition to other activities such as

tourism and fishing, all of which make the MUP a sustainable platform for harvesting food and energy with other activities.

Offshore platforms are not a new concept—they existed as early as the 19th century. The first offshore platform was for oil production was well-drilled in California by 1897. The first design for an offshore platform was in 1869 by Thomas Rowland, but it was never built because it was unrealistic idea at that time [2]. Oil and gas are the only mature sectors which have experience of constructing platforms further offshore. Thus, platform studies of oil and gas represent a large base for other sectors such as energy and aquaculture in terms of floating platforms and subsea engineering.

The objective of the present study was to review the proposed techniques and configurations of offshore platforms to give a complete view of the integrated offshore platforms in the EU. The paper is structured as follows: Section 2 gives a general idea about MUPs and highlight their components such as aquaculture and potential energy resources. Section 3 explores various structures of MUPs. Section 4 explains the design methodology for offshore platforms. Section 5 examines the electrical issues of MUP such as network configuration challenges. Sections 6 and 7 highlight the control of MUP grids and power quality challenges, respectively, while the last section concludes the study.

## 2. What is an Offshore Multi-Use Platform?

An offshore area can be defined based on different criteria such as distance from shore, water depth, jurisdictional boundaries, and wave exposure [2]. This offshore area is far from shore where there is a lack of topographical features such as capes, headlands or islands [3]. Some studies [3,4] have provided a classification for marine sites based on sea state or wave energy spectra; see Table 1. Class 1 refers to marine areas that are exposed to wave heights of less than 0.5 m, so the degree of exposure to strong waves is insignificant; these are normally the nearshore areas. The site class increases as it moves further from the shore, because it is more exposed to higher waves and more open ocean. When wave height goes above 3 m, sites are classified as extreme at site Class 5 and this is an offshore area. In between Class 1 and Class 5, there are various classes which used for classification of aquaculture cages techniques.

**Table 1.** Classification of offshore sites [4].

| Location Class | Wave Height | Exposure Degree to Wave |
|:---:|:---:|:---:|
| 5 | Higher than 3 m | Extreme (offshore area) |
| 1 | Below 0.5 m | Insignificant (Nearshore area) |

An MUP could be defined as an area of sea or ocean that combines different activities such as aquaculture, tourism, transportation, oil production, and energy farms. The combination of these activities could be completely integrated into one platform (shared structure) or could just share a marine space (shared area); more focus on this comes later in Section 3 [5]. Bringing different activities together could potentially benefit each other by lowering installation and maintenance costs, increasing resource utilization, reducing the environmental impact, etc. [6]. To sum up, an MUP comprises different kinds of renewable energies based on site parameters, in addition to other activities such as aquaculture, maintenance service, and leisure, as shown in Figure 1. The energy array in Figure 1 involves many hybrid energy units because every unit could comprise a wind turbine, a wave converter, and a small solar farm. Figure 1 illustrates the idea of an MUP, and, later, Section 3 shows that there are many other structures with different concepts. One of the ideas of MUPs is to use a floating substation that is separated from the platform that could export energy to the shore or supply onsite loads via a main distribution board (MDB).

The European Union (EU) oversees two big programs to back the concept of offshore MUPs. The Ocean of Tomorrow is a primary step to pave the road for the other bigger European project "Horizon 2020." The main goal of these projects is to develop a multi-use platform (MUP) in order to

extract energy from marine resources, as well as other uses such as aquaculture in the same area or structure. Horizon 2020 is a more recent program and is considered to be a continuation of "The Ocean of Tomorrow." It is the biggest funded program by the EU for research and innovation, ultimately costing 80 billion euros over seven years from 2014 to 2020. Some of the Horizon 2020 projects are provided in Table 2. The Ocean of Tomorrow project focused strongly on technologies and innovation issues for marine activities in ways that do not have negative impacts on the marine ecosystem. The Ocean of Tomorrow continued over the period of 2010 to 2013, and it comprised 31 projects under the framework program (FP7), as per Table 2, which shows some of these projects. For interested readers, the first book about offshore platforms was recently published by Koundouri [7]. It provides an environmental and socio-economic assessment of multi-use offshore platforms.

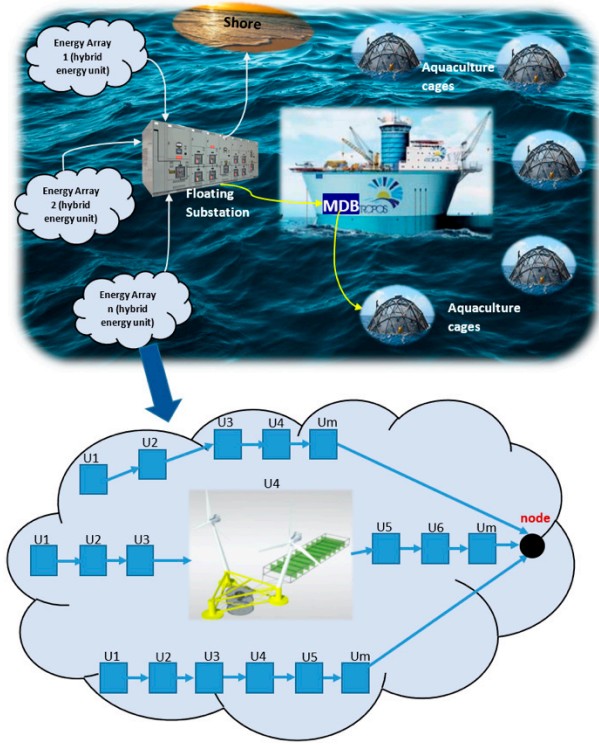

**Figure 1.** Schematic diagram of a potential multi-use platform (MUP).

**Table 2.** Ocean of Tomorrow and Horizon 2020 Projects.

| Project: The Ocean of Tomorrow [8] | EU Fund | Status |
|---|---|---|
| *FP7-Ocean-2010* | | |
| Arctic Climate Change Economy and Society | € 10,978,468 | Done |
| Vector of Change in Oceans and Seas Marine Life | € 12,484,835 | Done |
| Sub-seabed CO2 Storage: Impact on Marine Ecosystems | € 10,500,000 | Done |
| *FP7-Ocean-2011* | | |
| Development of a Wind-Wave Power Open-Sea Platform | € 4,525,934 | Done |
| Innovative Multi-Purpose Offshore Platforms: Planning, Design and Operation | € 5,483,411 | Done |

**Table 2.** *Cont.*

| Project: The Ocean of Tomorrow [8] | EU Fund | Status |
|---|---|---|
| Modular Multi-Use Deep Water Offshore platform | € 4,877,911 | Done |
| Marine Microbial Biodiversity, Bioinformatics and Biotechnology | € 8,987,491 | Done |
| *FP7-OCEAN-2012* | | |
| Priority Environmental Contaminants in Seafood: Safety Assessment, Impact and Public Perception | € 3,999,874 | Done |
| Integrated Biotechnological Solutions for Combating Marine Oil Spills | € 8,996,599 | Done |
| Suppression of underwater Noise Induced by Cavitation | € 2,999,972 | Done |
| Science and Technology Advancing Governance on Good Environmental Status | € 999,733 | Done |
| *FP7-OCEAN-2013* | | |
| Marine Environmental In-Situ Assessment and Monitoring Tool | € 5,434,221 | Done |
| Real-Time Monitoring of SEA Contaminants by an Autonomous Lab-On-A-Chip Biosensor | € 5,751,459 | Done |
| Sensing Toxicants In Marine Waters Makes Sense Using Biosensors | € 4,144,263 | Done |
| Marine Sensors for the 21st Century | € 5,924,945 | Done |
| Low-Toxic, Cost-Efficient, Environment-Friendly Antifouling Materials | € 7,447,584 | Done |
| Synergistic Fouling Control Technologies | € 7,995,161 | Done |
| Logistic Efficiencies and Naval architecture for Wind Installations with Novel Developments | € 9,986,231 | Done |
| **Project: Horizon 2020 [9]** | | |
| United Multi-Use Offshore Platforms Demonstrators for Boosting Cost-Effective and Eco-Friendly Production in Sustainable Marine Activities | € 11,399,118 | End 2023 |
| Lean Innovative Connected Vessels | € 7,808,691 | Done |
| Functional Platform for Open Sea Farm Installations of the Blue Growth Industry | € 9,854,077 | End 2021 |
| Multi-Use in European Seas | € 1,987,603 | Done |
| Multiple-Uses of Space for Island Clean Autonomy | € 9,834,521 | End 2024 |
| Multi-Use Affordable Standardized Floating Space@Sea | € 7,629,927 | End 2020 |
| Marine Investment for the Blue Economy | € 1,977,951 | Done |

### 2.1. Aquaculture

Aquaculture one of the most promising sectors for offshore platforms because it is an increasingly important contributor to economic growth and global food supply. Aquaculture is the fastest-growing animal food-producing sector on the planet. While capture fisheries production decreased by 2.6% from 1992 to 2012, this reduction was compensated for by the increased global supply which rose from 15% to 42% over the same two decades. China is the biggest producer of aquaculture products in the world, with around 62% of the world's fish and shellfish, while European aquaculture contributes less than 2% of the global aquaculture production in terms of weight [10]. This lack of growth in the EU's aquaculture could be explained by its strict environmental regulation, and this may open the door for more environmentally friendly alternatives such as integrated offshore farms. Offshore fish farming has been recognized as an alternative option for increasing seafood, and there has been international attention on this issue since the 1990s [11]. In its attempt to improve the aquaculture situation in Europe, the EU issued this policy statement: "Fish cages should be moved further from

the coast, and more research and development of offshore cage technology must be promoted to this end. Experience from outside the aquaculture sector, e.g., with oil platforms, may well feed into the aquaculture equipment sector, allowing for savings in the development costs of technologies" [12].

The relatively controlled and planned nature of aquaculture with respect to fishing guarantee, to a large extent, better social stability in coastal communities and achieves progress in job creation especially when integrated with other activities such as renewable energy, tourism, and conservation [13]. Moving to open-sea fish farming bears huge opportunities for this industry. The ocean has enough space for farms extensions, it has no or reduced conflict with user groups, and farms will be in a safe place because it is far from human sources of pollution. Using offshore farms would have a positive environmental impact while reducing the costal fish farms that have a negative impact on the environment. In addition, offshore farms will be in optimal environmental conditions for a wide variety of marine species [12]. Moving offshore would facilitate operations such as hydrolysing and thermalizing, which are highly important for salmon fish [13]. Offshore fish farms have the potential to develop and increase organic production [10]. At offshore sites, a greater water exchange makes it easy to remove farm waste and offers better salinity stability [14]. Furthermore, larger and deeper cages that are located further offshore would provide a safer environment for some European species, such as seabass and seabream [13]. Integrating aquaculture with offshore platforms including wind farms, aquatic sport centres, angling centres, and tourism facilities is a great potential business opportunity. Additionally, they would be very good academic centres for studying energy and aquatic animal lives.

Having said that, there are environmental concerns regarding the moving fish farms offshore such as nutrient and chemical pollution, habitat damage, disease introduction, and the interbreeding of wild stock with escapees from farms; the productivity of a farm depends on location choice, and so spatial planning is required to ensure that farms have no impact on their surrounding ecosystems and to ensure the sustainable growth of offshore aquaculture. In addition, offshore farms can be exposed to wild predators such as otters, sea lions, seals and birds. Additionally, there is a fear of local oxygen depletion due to the organic matter that is likely to fall to the seafloor during fed and unfed aquaculture operations. However, there are approaches to reduce such pollution. Lastly, there are higher costs for farm operations with offshore aquaculture.

To sum up, aquaculture from land-based and nearshore fish farms have experience a lot of critique and have many challenges due to economic issues, political issues, environmental issues, and resource limitations [11]. Moving aquaculture offshore is urgently necessary, but there are concerns that need to be addressed to ensure the sustainable growth and to overcome administrative and regulatory barriers such as licenses, spatial planning, the use of water resources, multi-level governance, and competition at fish markets [10].

*2.2. Potential Offshore Energy Resources*

Some studies have proposed the combining of marine energies as a better alternative instead of using a single source of energy [15–17]. They claim that many advantages could be achieved such as better liability systems, increased energy yields, smooth output power, and shared grid infrastructure. In addition, this option is environmentally friendly when compared with the independent installation of energies [17]. The costs of construction and maintenance could be significantly reduced via the use of shared resources such as foundations, logistics, operations, and maintenance. For example, the MWh generated from wave converters is still more expensive than their counterparts from other renewable sources and conventional sources of energy when constructed individually [18].

The main ocean energy technologies are the wave, tidal and ocean thermal energy conversion (OTEC). Wind and solar energies are not marine-based, but they could be implemented offshore at different scales. Today, ocean energy from various marine energy resources is developed to commercial scale with a generation capacity of 0.5–17 GW under construction. Tidal range energy predominates all forms of ocean energy in terms of installed capacity, with a rate of around 99% [19]. The following

sub-sections explore potential renewable sources of energy that could be implemented on offshore platforms such as wind, wave, solar, tidal, ocean thermal energy conversion, and biomass.

- Wind Energy Converter

There are two main categories of wind turbines: horizontal and vertical axis. The three bladed horizontal axis wind turbine (HAWT) is the most popular wind turbine that is used offshore and onshore in a commercial way [20]. On the other hand, vertical axis wind turbines (VAWTs) have the advantages of simpler structures that make them a better option for floating turbines and cut costs of foundations and quicker response for changing wind direction. Moreover, VAWTs offer lower noise levels, which makes them suitable for MUPs. They have simpler control system as there is no need for pitch control, and their simple stricter means that any required maintenance is easier to perform [20]. Many recent studies have developed VAWTs [21–23]. In this regard, the authors of [23] proposed a new unusual design for a wind turbine that could reduce the cost by 65%.

- Wave Energy Converter

Abundant energy could be harvested from the ocean, along with wind energy. The existing wave turbines can be divided into two main groups, the first of which is direct action turbines that directly convert hydrodynamic energy into electrical energy. The indirect group does the same function indirectly. The first group of turbines has a simpler structure and, as such, is more reliable and less costly [23]. It is worth mentioning that the first open sea testing facility at Orkney Island in Scotland (in operation since 2003) has a wave testing site. It comprises five berths of 2.2 MW power capacity. Another grid-connected wave hub in South West England includes four separate berths, each with a capacity of 4–5 MW [24]. Recently, the NEMOS team developed a wave energy converter to be installed offshore. This converter has an $8 \times 2$ m-sized floater and a structure that is 16 m long. The team is currently working on the installation of a large-scale prototype in the North Sea that could generate enough energy for several households. The standalone floating design of this converter makes it ideal for offshore installation with an MUP [25].

- Tidal Energy Converter

There are many ways to classify tidal turbines. The most common category is based on conversion technology. The authors of [26] explored the most popular types of tidal turbines based on conversion technology. Other studies, such as [27–29], have classified hydrokinetic turbines in more detail. Real projects include the Skerries Tidal Energy Array with 10 MW in Wales, which has been in operation since 2015, and the Irish Open Hydro Tidal Energy Array project, which is bigger at 100 MW and is expected to be in commercial operation by 2020. Moreover, the first open sea testing facility on Orkney Island in Scotland has a tidal testing site. It is located near Eday at water depths of 12 and 50 m on an area of $2 \times 3.5$ km. It has eight berths of 5 MW power capacity [24]. Tidal energy is included in this study with expectations that the floating version of the tidal turbine (SR2000) will be developed for offshore installation in the future, though such a version is limited to a water depth of 25 m so far.

- Floating Solar Farms

Floating Photovoltaic (PV) panels have been proposed under the Modular Multi-use Deep Water Offshore Platform Harnessing and Servicing Mediterranean, Subtropical and Tropical Marine and Maritime Resources (TROPOS) project as potential offshore energy sources. Floating PV panels have proven themselves as an available technology that is already used in onshore lakes, reservoirs and ponds. There is a floating PV farm at Far Niente Winery, USA. There is a floating PV farm under construction in Ichihara city in Japan with a power capacity of 13.4 MW [30]. However, due to the harsh marine environment, there are no PV structures installed offshore in open water. PV panels should be qualified to withstand humidity, salinity, sea spray, corrosion, and fouling in the open seas. The use of PV panels in a sea environment is still very limited, and, so far, they have only been used on boats or as hybrids with a wind turbine in a pilot project [31].

- Ocean Thermal Energy Converter (OTEC)

The extraction of energy by using an OTEC is based on the thermodynamic Rankine cycle which is used in steam power plant [32]. For an OTEC to be a feasible source of energy, a minimum temperature difference between the warm water and water at 1000 m deep should be at least 20 °C [33]. For this reason, harvesting this energy is specifically available in tropical areas. An OTEC is classified based on the location of the plant or the thermodynamic cycle used. For location-based units, there are three kinds of plants: floating, self-mounted and land-based. In terms of the thermodynamic cycle, there are three main types: open-cycle, closed-cycle, and hybrid-cycle. For interested readers, details about all types have been given in the ocean engineering handbook. Electricity and fresh water are the main outputs of a large OTEC plant, but various by-products could be harvested such as air-conditioning and aquaculture [32]. One 50 MW hybrid cycle plant could provide the daily water needs for a small community with 300,000 people. In addition, deep water is 28 times richer in inorganic nutrients such as nitrates, silicates, and phosphates, which could be used in a commercial way for sea farming.

- Marine Biomass Energy

Conventional technologies that are used for extracting biofuels are based on animal oils, vegetables, starch, and sugar, but these methods have been widely criticized because they consume food resources. For this reason, marine algae have appeared on the horizon as more environmentally sustainable and friendly feedstock because they do not compete with food resources, they save freshwater, and they could be grown by wastewater. Algal feedstock could be used to obtain energy and non-energy products. Many biofuels could be extracted from algae such as biodiesel, bioethanol, biogas, and bio-jet fuels [34]. Bharathiraja et al. [35] concluded that biofuels from marine algae are still not economically feasible, an issue which comes back to the high costs of the operation, cultivation, processing, and separation of biofuels. However, integrating algae farms in the offshore platform proposed under this study would improve the technology and make it available at reasonable costs. In addition, depending on algae, biofuels that are used as potential storage can provide better reliability for offshore islands and can avoid the use of large capacities of batteries.

## 3. Offshore MUP Structures

Combining marine resources of energy (wind, wave, tidal, floating solar farms, algae biomass, and ocean thermal energy conversion) with the different activities mentioned above could be fulfilled in very different ways and concepts. Various structures have been proposed under the funded projects of the EU. These projects are Innovative Multi-purpose offshore platforms: planning, Design and operation (MERMAID), Development of a wind-wave power open-sea platform equipped for hydrogen generation with support for multiple users of energy (H2Ocean) and TROPOS. Offshore wind energy appears strongly in all structures because it is an already developed and mature energy. Different wave converters share these structures to promote wave energy, which still at an early stage of development. The combination of wave and offshore wind energy to generate electricity is a recent topic, and few studies have tackled this issue. Most works in this regard have been done by EU-funded research projects, as mentioned earlier. Khrisanov et al. [23] highlighted the advantages of using the hybrid wave/wind power system; they found that hybrid floating wind and wave power systems are promising directions for more harnessing ocean energies. There are two main concepts for hybridizing these two sources of energy: mechanical and electrical combination. The first is combining wind and wave turbines in a mechanical complex system, and the resulting rotation moment of both turbines is used to drive the generator's rotor. Unfortunately, this kind of system suffers from less reliability and increased costs [23]. Thus, this kind of system has not been used, and an electrical hybrid system has been proposed. The electrical system depends on the electrical combination between the wind and wave converters, i.e., each converter has its own generator and the output power of both are combined via a power electronic converter.

There is a general classification for the offshore structures in terms of foundation type as a function of water depth: Fixed structures are constructed in shallow water with water depth less than 50 m, and floating structures exist in water at a depth of larger than 50 m [24]. An MUP could be classified based on its technological basis, its relative location to the shoreline, or its water depth. However, this study classifies platforms to three categories based on the connectivity among activities to co-located systems, combined structures, and island structures.

### 3.1. Co-Located System

Wind farms and wave arrays share the same marine area, maintenance, operation equipment, activities, grid connections, etc., thought they do not share foundations (See Figure 2). This kind of combination is proposed for the early stage of development.

### 3.2. Combined Structure

The idea of this structure is highlighted under the MERMAID project. In these structures, different energy converters share the same foundation and connections, and everything is shared as a unit (see Figures 3–15). Some loads such as aquaculture and algae farms could be attached to that unit, as proposed in the TROPOS project (Satellite Unit; see Figure 4). The foundation of this structure could be bottom-fixed or floating as per Figure 3a,b, respectively [17].

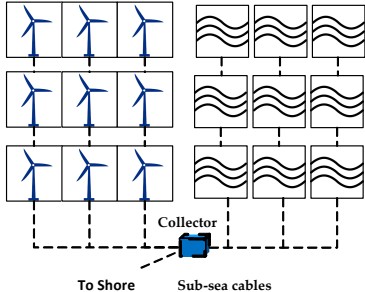

**Figure 2.** Co-located independent array.

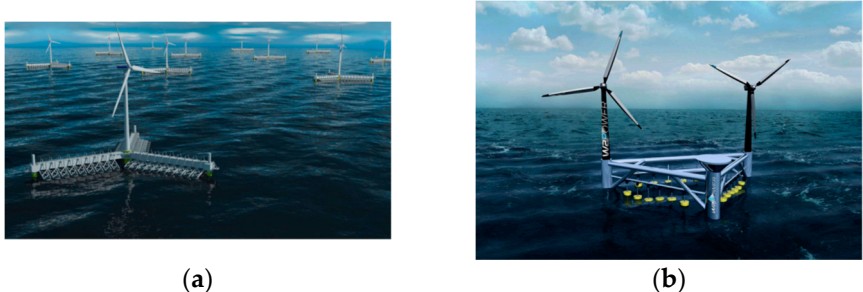

(**a**)　　　　　　　　　　　　　　　(**b**)

**Figure 3.** Hybrid system. (**a**) Fixed bottom. (**b**) Floating system [24].

- Satellite Unit Structure

The TROPOS project proposed what is called a floating satellite unit (see Figure 4), which combines wind turbines, PV solar panels, and an aquaculture breeding fish facility with an algae farm attached to it [36].

- Poseidon Wave/Wind Structure

The platform proposed under the MERMAID project that was developed by the Poseidon Floating Power Company with a floating foundation and a combination of wind and wave converters, as shown in Figure 5 [24].

- Two Wave One Wind Structure

This structure was proposed by the MERMAID project and developed by Ocean Wave and Wind Energy Company with a fixed foundation. It combines a wind converter with dragon and point absorber wave converters, as shown in Figure 6 [24].

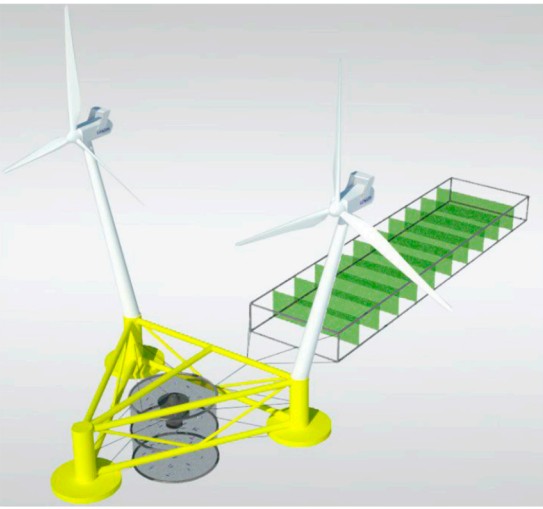

**Figure 4.** Satellite unit proposed in green and blue concept [36].

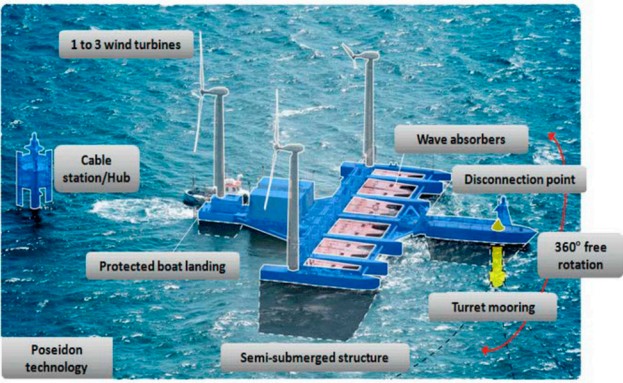

**Figure 5.** Poseidon wave/wind energy platform [24].

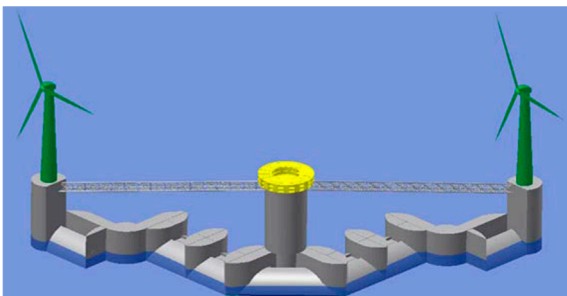

**Figure 6.** Two wave structures and one wind structure [24].

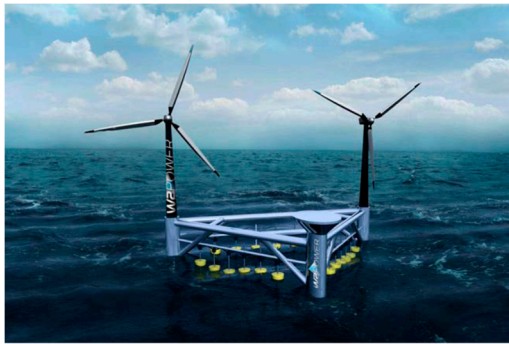

**Figure 7.** Two wind Power structure [24].

- W2Power Structure

This was proposed under the MERMAID project and was developed by the Pelagic Power Company with a semi-submersible floating platform. It combines two wind turbines with a wave converter (point absorber type), as shown in Figure 7 [24].

- Wave Treader

This was proposed under the MERMAID project and was developed by the Green Ocean Energy Company. It has fixed piles and combines a wind turbine with a wave converter, as shown in Figure 8 [24].

- Seagen W-Shape

This was developed by the Seagen Company. It has a fixed pile (monopole) and combines a wind turbine with a tidal converter, as shown in Figure 9 [24].

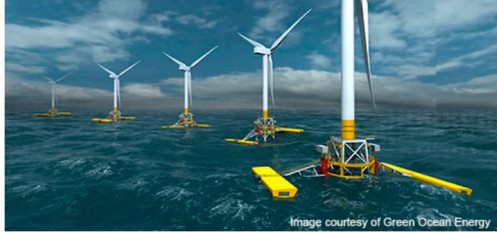

**Figure 8.** Wave Treader Structurer [24].

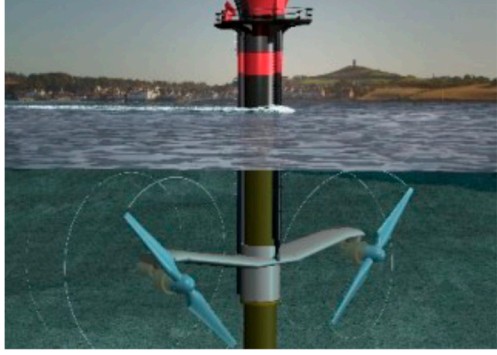

**Figure 9.** Seagen W structure [24].

- WEGA: Hybrid Coupling (Wind turbine + WEC) Structure

This was developed by Sea for Life Lda. It has a fixed foundation and combines a wind turbine with a Wave Energy Converter (WEC), and other uses could be added, as shown in Figure 10 [24].

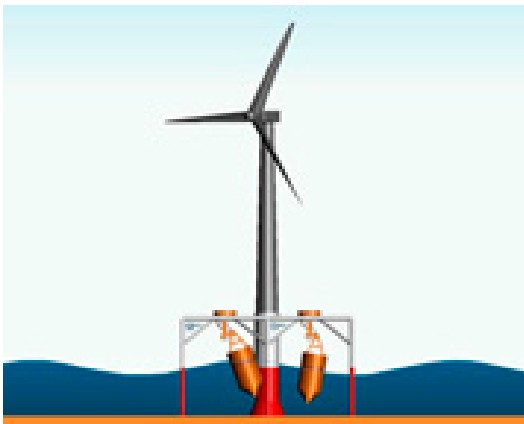

**Figure 10.** WEGA Hybrid Coupling (Wind turbine + WEC) structure [24].

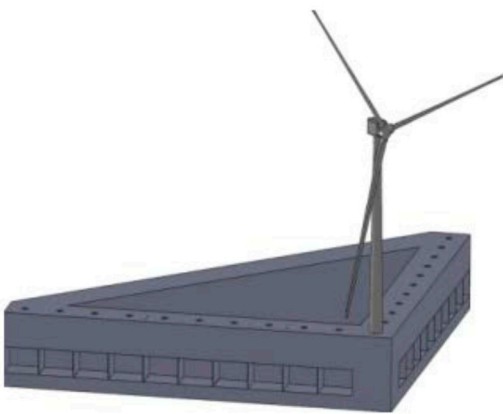

**Figure 11.** Triangular structure [24].

- Triangular Structure

This structure was proposed and developed under the Marina platform project. It has a semi-submersible floating foundation and combines a wind turbine with a wave converter, as shown in Figure 11 [24].

- Three Branches Wave/Wind Structure

This structure was also developed and tested under the Marina platform project. It has a semi-submersible floating platform and combines a wind turbine with a wave converter (Flabs type), as shown in Figure 12 [24].

- Spar Floating Buoy Structure

As with the previous two structures, this structure was developed and tested under the Marina platform project. It has a spar floating buoy and combines a wind turbine with a wave converter (point absorber type), as shown in Figure 13 [24].

- Cantabria Platform Structure

This structure was developed and tested in a laboratory at the Cantabria site to validate the final design. The semi-submersible floating platform has three oscillating water columns (OWCs)

constructed of two wave converters and one wind turbine, as shown in Figure 14. The power capacity of the proposed floating structure is 8 MW (5 MW from the wind turbine and three OWCs with 1150 KW each) [24].

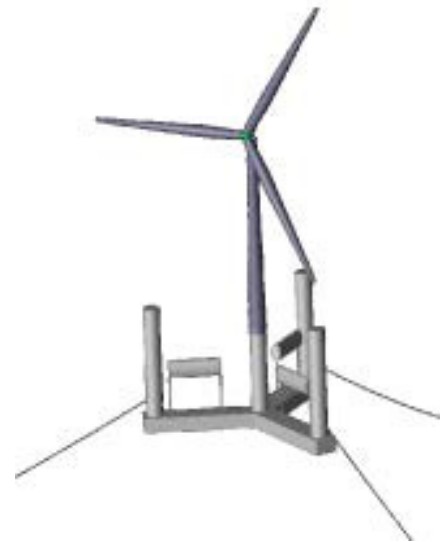

**Figure 12.** Three branches wave/wind structure [24].

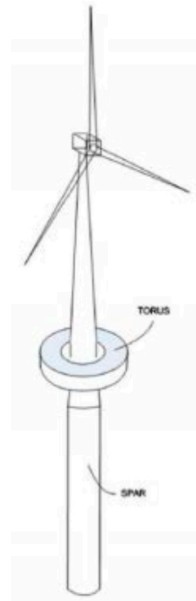

**Figure 13.** Spar floating buoy structure [24].

- Hoxicon Platforms

This was developed by the Hoxicon Company. There are various shapes of the floating platform, which only combines wind turbines, as shown in Figure 15 [24].

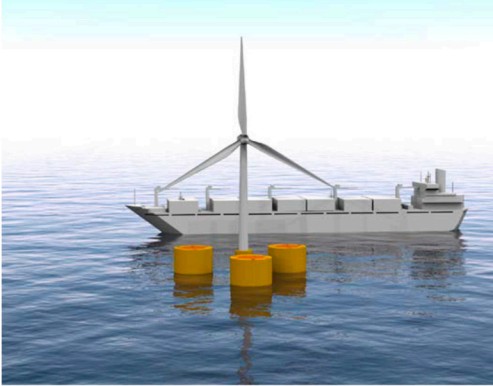

**Figure 14.** Platform at Cantabria, MERMAID project [24].

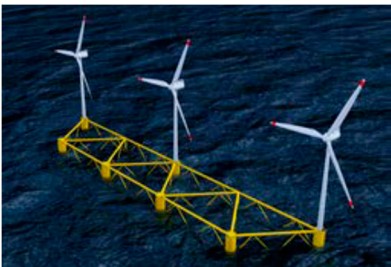

**Figure 15.** Floating structure that combines wind turbines [24].

### 3.3. Island Structure

The concept of an island structure is becoming very clear under the TROPOS project via the three proposed island configurations, which are the sustainable service hub island, the green and blue island, and the leisure island. TROPOS has aimed to develop a floating MUP to adapt to deep waters with focus on Mediterranean, tropical and sub-tropical areas [37]. Basically, the four main sectors to be integrated into these islands are those of transport, energy, aquaculture and leisure (TEAL). For more information on two kinds of islands—artificial and floating islands—interested readers can look at [17].

- Sustainable Service Hub Island

This is a floating offshore platform of an industrial nature, as it includes a lot of cranes and workshops. This concept focuses mainly on energy and transport issues, though it still has leisure activities and aquaculture (see Figure 16). It includes a large floating offshore port with repair and maintenance facilities for large ships. Lifting capabilities have been proposed for workshop activities and for material storage and handling. The deliverable D4.3 "Complete Design Specification of 3 References TROPOS systems" explores all required elements on this platform [36].

Based on this proposed platform, TROPOS has had big influences on the transport, energy and aquaculture sectors in terms of reducing the operation and maintenance costs that are related to these sectors. The implementation of other energy sources such as solar (photovoltaic or thermal), OTECs, and marine energies within the same platform can all act to reduce related costs and increase the reliability of the power system. This platform could serve the transport and mining industries by providing them with fuel, electrical energy, food and freshwater.

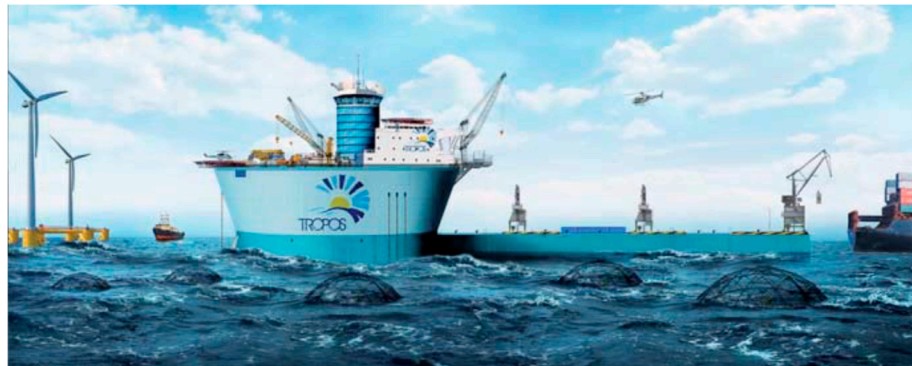

**Figure 16.** Configuration of the industrial complex concept [38].

Within this configuration, energy production is comprised huge renewable sources of large wave and wind farms. Wind turbines could be integrated with the platform itself or with floating turbines. The idea of this configuration is that the generated energy supplies all facilities at the platform, while the excess energy is used to produce gases and liquid fuels that are stored as energy storage to be used as a fuel for fuel cells or even internal combustion engines.

There have been few case studies for this concept, which are used to perform maintenance operations for the existing offshore wind farms. Horns Rev2 was established in the Danish North Sea to provide service for a 209 MW wind farm. Nordsee Ost, Dan Tysk and MittlePlate are other three case studies which were constructed in the German North Sea for maintenance purposes as well [39].

Aquaculture in this platform is limited because it conflicts with the other considered facilities, such as workshops and material handling. Thus, when possible floating cages could be used for aquaculture in different locations, such as areas between wind turbines and areas that are close to the platform. In this case, feeding operation could be managed via the platform itself or from independent floating silos among the cages [38].

It is worth mentioning that this configuration and the green and blue configuration are designed to be grid-connected. However, grid-connected offshore floating wind turbines and arrays have yet to be developed due to a lack of experience. An Edinburgh University report [40] suggested that a mobile floating offshore substation could be used until a floating substation is proved.

- Green and Blue Island

This concept is mainly focused on physical and biological ocean resources to extract food and energy [36]. An offshore wind farm, a potential wave energy farm, and an OTEC, if applicable with aquaculture installation, are considered to be the main contents of this configuration (see Figure 17). It has been proposed that aquaculture that is integrated into the same floating platform of other activities is called a shared structure. Other than the previous configuration, this one completely avoids industrial activities that might jeopardize the aquaculture sector. There are two main sub-concepts for this configuration: One is wind and wave plus aquaculture, and the other is aquaculture with an OTEC [38]. Configuration details about the above two sub-concepts and other configurations are available in [38,41] and the energy island website [42].

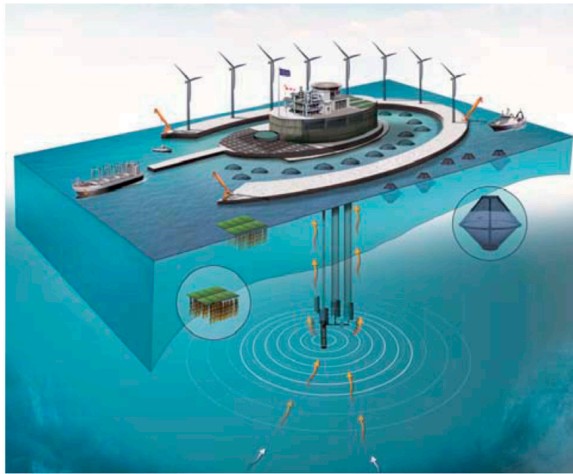

**Figure 17.** Green and blue platform configuration [38].

This configuration comprises two main structures: the central unit and the floating module. The central unit involves various sections such as the crew's accommodation, a rescue system, communication, electrical units, a fish processing plant with all required areas, a unit for exporting aquaculture, and laboratories for aquaculture facilities. On the other hand, the floating unit has areas for twenty-foot equivalent unit (TEU) of storage and satellite spare parts, as well as berthing capability for an offshore supply vessel.

The green and blue concept focuses on algae as a source of energy by converting biomass to energy. Algae could be used as biomass to harvest energy and non-energy products. From algae, energy products such as biodiesel, biogas, bioethanol, and bio-jet fuels are produced. On the other hand, algae produce non-energy products such as carbohydrates, pigments, proteins, biomaterials, and bioproducts [34]. The algae farm is attached to a satellite unit (see Figure 4). This unit has two wind turbines which are integrated into a floating structure with a fish cage and a floating algae farm [36]. It is worth mentioning that the different kinds of generators, integrated into this configuration, would be operated based on the fuels that are produced from the algae farm to get a fully sustainable system.

- Leisure Island

This platform is in relatively shallow water near the coast, compared to the other previously mentioned configurations. It includes different modules: a diving centre, an aquaculture structure, a water sports centre, and an underwater observatory to watch the marine environment and aquaculture around the site (see Figure 18). The floating module in this configuration is a bit different from the other two configurations because it has a PV plant with storage and a substation to provide electricity to the central module when required [36].

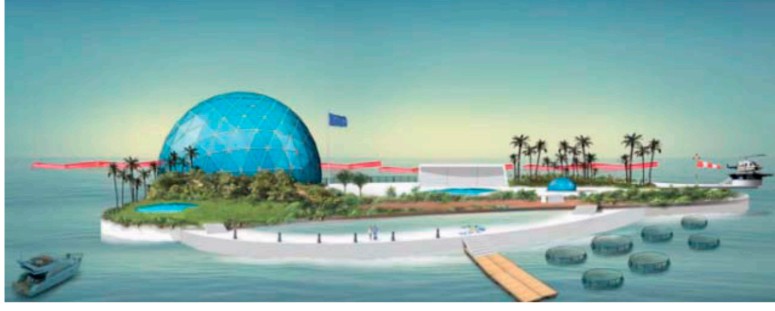

**Figure 18.** Leisure island configuration [38].

The island should be self-sufficient in terms of all kinds of energy demands: electrical power, hot water, and air conditioning inside the buildings. For this purpose, solar energy—either PV or thermal with storage—is extensively integrated into the island's architecture, and wind turbines are integrated into this configuration. In other words, the energy demand for this configuration is met by wind and solar energy [38].

Though an MUP can provide sustainable and economical solutions for the problem of a lack of seafood and higher prices of offshore energy, it should be designed in a way to avoid or reduce negative environmental and ecological impacts. Some stakeholders have concerns about living marine environments and habitats that could be affected by foundations of MUPs and cages [5]. Additionally, there are concerns about MUPs conflicting with other marine activities such as transport, tourism, fishing, entrance to marine ports, and wildlife and birds area protection. Marine litter is another problem that could be increased with commercialized MUPs. Plastic alone (around 60–80% of marine litter) was estimated to exist as 275 million metric tons (mt) in 2010, and this quantity could be increased by increasing the number of MUPs. Marine litter has a negative impact on human health, marine environments, marine ecosystems, marine industries, and marine species, and this leads to negative economic impacts. Another challenge for the energy system of MUPs is that they should use 100% renewable energy resources to avoid releasing $CO_2$, the use of which leads to ocean acidification, which has a negative impact on marine ecosystems [43].

## 4. Offshore MUP Design Methodology

Designing offshore platform requires the assessment of the project site from different sides: technical, economic, social and environmental. Thus, involving all relevant stakeholders at an early stage of development is required because the design of such installations depends on experts' judgement from different sectors. Barbara et al. [44] proposed a methodology consisting of four phases for the purpose of design of an offshore platform: the pre-screening phase, the preliminary design of the single-use platform, the ranking phase, and, lastly, the preliminary design for the selected multi-use platform, as shown in Figure 19.

The pre-screening phase examines the platform components (energy sources, aquaculture, marine service hub, and leisure island) based on the site conditions in terms of wind speed, yearly wave power, tidal range, and potential fish production. The outcome of this phase is to define the various uses that are integrated onto the offshore platform at a specific site. The preliminary design phase chooses the most suitable energy converters and applicable fish farms based on the site assessment that was accomplished in the first phase and took legal constraints into account [44].

The third phase is a ranking step to give a score for each component in the platform based on different aspects such as the technology development level in terms of reliability and performance, the installation and maintenance costs of various elements as a function of system mechanical complexity and water depth, and potential risks in terms of pollution, power take-off failure, geotechnical failure, and structure modularity.

Then, the assigned scores for each module are combined and alternatives are weighted based on the score of each combined module. The outcome of this phase is the determination of the best scheme for an offshore platform at a specific site. The last phase is the preliminary design of the selected scheme, taking the interaction and the conflict between different modules on the platform into account, in addition to the optimization of spatial planning [44]. For example, the proposed service hub platform in the TROPOS project has a conflict between the aquaculture cages and the service activities make surrounding water not pure enough for growing fish.

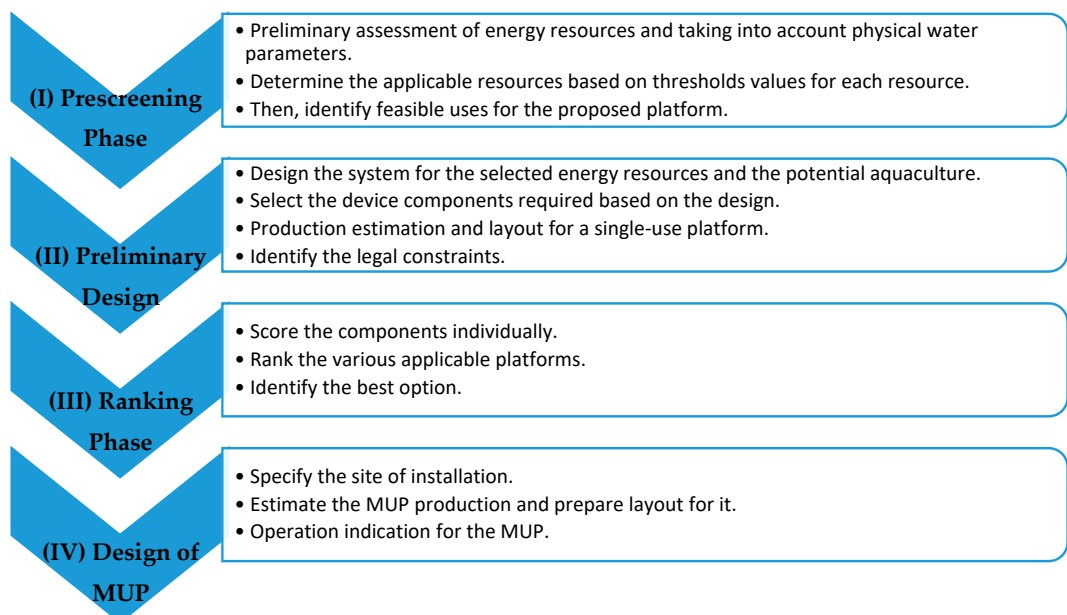

**Figure 19.** Offshore platform methodology phase [44].

## 5. Offshore MUP Grid Configuration

This section highlights the electrical connection between different components of an offshore platform. It is important to differentiate between two different electrical layouts: the hybrid energy resources layout and the MUP local grid layout. Hybrid energy resources represent the generation station or the supply to the MUP local grid. This supply depends on different marine energies, as presented in Section 3, of the combined structures. To the authors' knowledge, the electrical connection between such structures has not been reported in the literature of MUPs. For this reason, this paper proposes the layout of wind energy farms to be applied for hybrid energy resources that considers a structure as one unit, as shown in Figure 20. A cluster within the wind farm could include *n* wind turbines, but under this study, it combines *n* structures when a structure could have various turbines. Figure 20 shows the connection levels, starting from the structure level, through the cluster and array level, to the node level, and ending with the substation level. A substation could supply the offshore platform or export energy, after adding extra equipment, to the shore in the case of large-scale energy farms.

A node is a single collecting point within an array [45]. An array connection could be a proper alternative when multiple devices are connected together in an array. There are different array schemes that are chosen based on geotechnical conditions and resource characteristics. The size of an array is limited by the acceptable voltage drop along cables and the array's maximum capacity [24].

An offshore MUP local network has been explored under the TROPOS project for the three different island configurations presented in Section 3.3. Two Alternating Current (AC) low voltage levels have been introduced: 400 VAC, which supply loads for sections such as those of acclimatization, a refrigeration system, a lubricant system, a sewage system and a rescue system, and 230 VAC, which supplies sections such as those of the aquaculture and algae systems, illumination, restaurant, hotel, and battery charger, in addition to a 24 VDC line for DC loads (see Figure 21). The daily load profile of the three island scenarios is provided in [36].

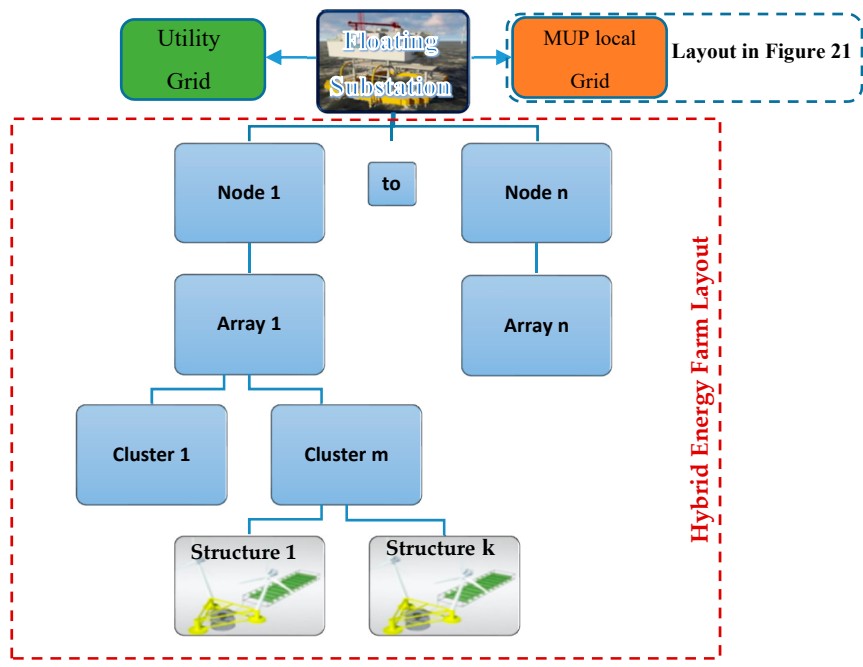

**Figure 20.** Configuration of hybrid marine farm supplies an MUP's local grid.

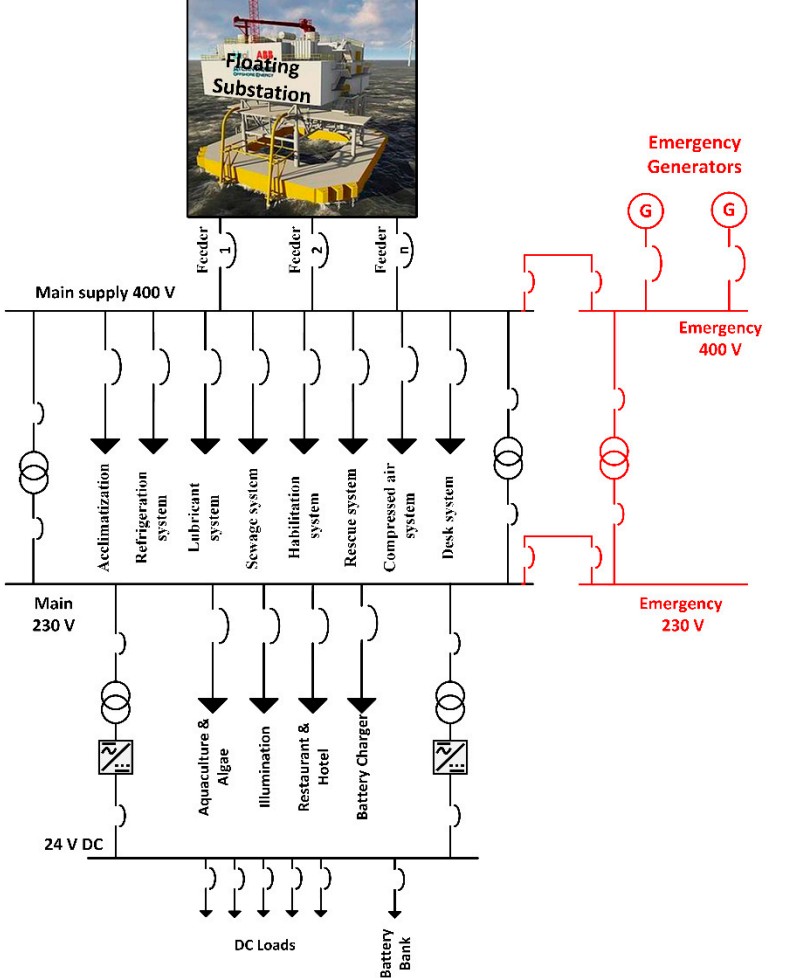

**Figure 21.** Single line diagram for an MUP's local network.

It is worth mentioning that the MUP local network presented in Figure 21 was modified from that of the TROPOS project. An MUP local grid of an island structure under TROPOS is supplied by generator sets and from the utility grid. This conception does not make sense, because MUPs are designed for offshore areas and to operate in sustainable way. For this reason, Figure 21 shows an MUP that was modified to be supplied from a hybrid marine farm via a floating substation.

## 6. Offshore MUP Grid Control

Using on large synchronous generators in a conventional grid makes an MUP's control system simpler with respect to isolated grids. For example, a synchronous generator changes its output power in response to load change without the need for any control or communication links [46]. On the other hand, an MUP grid has a completely different nature from that of a conventional grid, as the former basically depends on a collection of inverters, synchronous generators, and asynchronous generators [24]. An MUP control system is required for the regulation of frequency and voltage, as well as for controlling load sharing among included micro-resources. In addition, it is necessary to resynchronize an MUP network with the main grid and power flow control between the two grids [47].

Some studies [46–49] have proposed a hierarchical control strategy with three control levels for controlling a microgrid by considering the islanded mode, the grid-connected mode, or connections with other microgrids; see Figure 22. Hierarchical control includes three control levels: a local controller, a central controller, and a supervisory controller from the lower to higher levels, respectively.

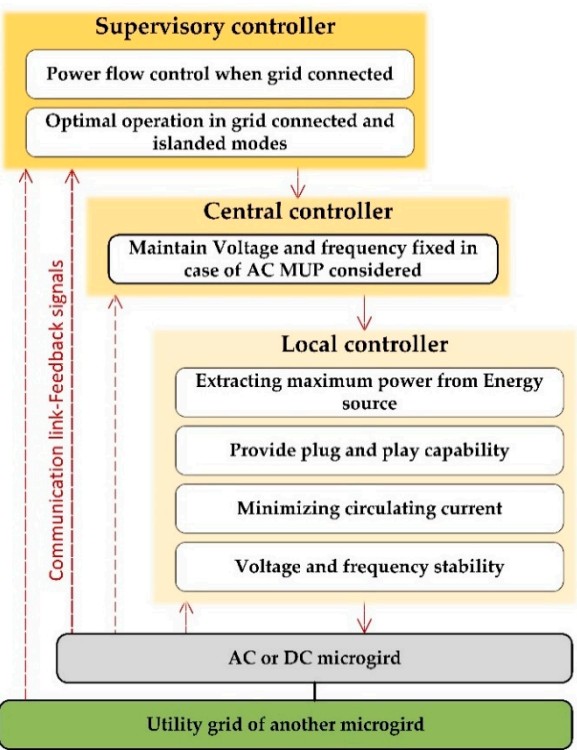

**Figure 22.** Microgrid Hierarchical Control level [46].

There are two kinds of local controllers: a micro source controller (MC) and a load controller (LC). An MC performs some local functions, such as controlling the voltage and frequency of microgrid in transient conditions, and it follows a Central Controller (CC) when connected to the grid. Both MC and CC are used to optimize the active and reactive power of the microsource and track the load after islanding operation. An LC is installed at the load side in order to manipulate the load via a central controller for load management [48]. There are various strategies for implementing local controllers, and these were well-presented in [46]. Frequency and voltage changes could occur with

a local controller even during steady-state. Thus, a central controller is used in order to compensate for this deviation. However, controllers at this level are designed with slower response time than local controllers. This is the slowest control level which manages the flow of power among microgrid and utility grid in order to achieve optimal economic operation [46].

## 7. Offshore MUP Grid Challenges

A local MUP network is very similar to a microgrid in terms of grid contents. The microgrid has a distribution system, micro-sources, loads, and a control system. An MUP grid is expected to comprise similar components, and it could be operated in an islanded or grid-connected mode, similarly to a microgrid. Moreover, an MUP network is based on many distributed generators (wind, solar, wave) that are unreliable sources of energy. Thus, an MUP network is anticipated to face similar challenges as a microgrid.

Generally speaking, a utility power grid basically depends on very large generator capacities that make it stable even with big disturbances occur. The situation is different with an MUP grid, which depends on many micro-sources to supply its fluctuating power. Due to the individual nature of the offshore grid, one of the challenges it could face is power quality problems. The authors of [50] identified these problems as shown in Figure 23.

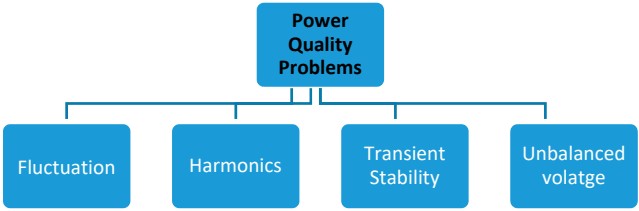

**Figure 23.** Power quality problems in the offshore local grid.

An offshore grid depends mainly on renewable sources of energy, such as solar or wind, that already have a low degree of controllability due to their output depending on the availability of resources. This could result in the output power of renewable energy systems being unsmoothed. To avoid this, Rashad [50] proposed a fuzzy logic controller as a way for mitigating the fluctuation of the output power of the wind turbine. Harmonics could arise due to electronic interfacing and nonlinear loads [50]. Harmonic frequencies have a negative impact on grid components, and this leads to reducing a system's lifetime, efficiency and reliability [51]. The harmonic current problem can be solved via a series active power filter (SAPF) [52] or a unified power quality conditioner (UPQC) [53]. Transient stability is another challenge for MUP grids, as it depends on small inertia generators that could suffer from even small disturbances. For this issue, battery storage has been used to support microgrids during transient conditions, and this strategy has shown better performance [54]. An MUP grid could suffer from an unbalanced phase voltage problem because it has many single-phase loads that are unbalanced in nature [50]. Various control schemes have been presented for voltage balancing in a microgrid [49,50].

## 8. Conclusion

The EU highly support the idea of MUPs for harvesting food and energy in a sustainable way over two big projects: the Ocean of Tomorrow and Horizon 2020. An MUP would be a good opportunity for the aquaculture sector because moving to open sea would provide enough space for fish farms, and they would be safe from human sources of pollution. In addition, the organic production of fish would be developed and increased. Various marine activities such as offshore energy, tourism, transport, and fisheries, would be positively influenced by MUPs, as many activities could be integrated together to get benefit from each other. Workshops being integrated onto MUPs would have a positive impact on transport, energy and aquaculture sectors, as the operation and maintenance costs of these sectors would be reduced. However, exploiting oceans and seas in a sustainable way still an environmental

and ecological challenge. There are concerns about the living marine environment and the habitats that could be negatively impacted by the foundation of MUP and fish cages. An MUP would conflict with other marine activities such as entrances to marine port and wildlife and birds area protection. Additionally, there is concern regarding increasing marine litter and ocean acidification.

This study had proposed classifications for offshore structures based on connectivity among different energy converters and activities of co-located systems, combined structures, and island structures. The island structure represents state-of-the-art offshore structures that have been proposed under the TROPOS project. The design methodology of MUPs has been explored. The pre-screening phase of this methodology is highly important for figuring out various activities that should be integrated into MUP-based on-site assessments for extracting the maximum benefit.

There is scarce literature about electrical grid configurations and control under MUPs. This paper has introduced a configuration of offshore local networks that in line with the control systems that are based on isolated microgrid literature for the above reason. Future studies of electrical local networks of MUPs should address challenges such as space limitation, the high costs of system components and installation, and a lack of available backup sources because MUPs are far offshore and use critical loads, such as aquaculture cages. A DC system has never been considered for offshore platforms, but such systems could be better alternatives when a simpler control system, lower costs and distribution losses, and increased system efficiency are required, so studying the feasibility of using DC systems for multi-use platforms is an open research area. In addition, algae biofuels would play a vital role because potential energy storage provides better reliability for offshore power systems and avoids using large capacities of batteries.

**Author Contributions:** Conceptualization, W.M.N. and O.A.-L.; Data curation, K.H.A.; Writing—Original Draft Preparation, W.M.N.; Writing—Review and Editing, M.E., D.C.-G.; Supervision O.A.-L.; All authors read and approved the revised manuscript.

**Funding:** This research received no external funding.

**Acknowledgments:** The authors would like to thank Ross Mackay for his efforts in proofreading of the manuscript.

**Conflicts of Interest:** The authors declare no conflict of interest.

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
