# Peer review of "Assessment of Multi-Use Offshore Platforms: Structure Classification and Design Challenges"

_sustainability, doi:10.3390/su12051860_

Round 1

Reviewer 1 Report

It is well known that the offshore areas represent a promising source of energy, being defined by many opportunities and challenges. The topic covered by the authors is interesting, aiming to provide a complete picture of the benefits that may be obtained from the development of some multi-use projects. Although this is a review paper, I consider that the authors fail to present a complete picture of this topic, and therefore in the current form the present manuscript cannot be considered for publication. Bellow I attach my points of criticism.

The subject of this work is more suitable for the topic of Journal of Marine Science and Engineering, which is another MDPI journal; Please discuss in the introduction, how a MUP project can be considered a sustainable project; Please define more clearly what an offshore area is, because this is the topic of this work. It seems that the authors cover an entire spectrum of marine applications, and easily jump from nearshore applications (tidal energy) to offshore ones (ex: wave energy); What is the relevance of the Table 1 for the present work? Also please discuss the info provided in this table, such as: what is a site class, how the wave heights were sorted, how does the degree of exposure used in aquaculture can be used for an MUP project; Please separate your results in two parts: a) what is real and developed at this moment; b) projects expected to occur in the near future. These results can be summarised in a table, in which you can indicate the status of each project. Regarding the wave energy, please consider to have a look on the NEMOS project; Section 5, 6 and 7 – cover some general aspects related to the grid connection. Please indicate more clearly what are the novelty provided in this section, because on a first look it seems that there are some basic aspects that need to be taken into account for all the offshore systems. There is a difference between the layout of an offshore wind farm and a MUP project? Please discuss these issues; Conclusion section – is to general. Please highlight loud and clear the main findings of the present work and briefly mention the challenges and benefits that are related to the development of a MUP project; Some references are missing in the text (ex: pg. 5/line 175 or pg. 11/line 374).

Reviewer 2 Report

In general, I miss a general critic overview about each reviewed issue, confronting the different alternatives, their well-known results (if they exist), pros and cons, etc. E.g., you talk about aquacultures at section 2.1. using a "triumphant" mood despite the general crisis that is devasting this productive sector and the very negative ecological effects on the marine ecosystems.

I miss too a review about the potential ecological effects of all the configurations, at least at a general level.

Please, try to include this critic point of view over the text in order to make possible that the readers can get a non dogmatic idea about this subject

Round 2

Reviewer 1 Report

Dear Sirs,

The manuscript is significantly improved and the authors answered to all my questions. From my point of view, the paper is suitable for publication.

Regards,

Reviewer

Reviewer 2 Report

No more comments to authors